# Learning Attention from Multi-Modal Imaging and Text: Application for Lesion Localisation in DWI

**Aydan Gasimova, Liang Chen, Paul Bentley, Giovanni Montana[*], Daniel Rueckert**
Imperial College London, [*]University of Warwick

## Abstract

Strokes are one of the leading causes of death and disability in the UK. There are two main types of stroke: ischemic and hemorrhagic, with the majority of stroke patients suffering from the former. During an ischemic stroke, parts of the brain lose blood supply, and if not treated immediately, can lead to irreversible tissue damage and even death. Ischemic lesions can be detected by diffusion weighted magnetic resonance imaging (DWI), but localising and quantifying these lesions can be a time consuming task for clinicians. Work has already been done in training neural networks to segment these lesions, but these frameworks require a large amount of manually segmented 3D images, which are very time consuming to create. We instead propose to use past examinations of stroke patients which consist of DWIs, corresponding radiological reports and diagnoses in order to develop a learning framework capable of localising lesions. This is motivated by the fact that the reports summarise the presence, type and location of the ischemic lesion for each patient, and thereby provide more context than a single diagnostic label. Acute lesions prediction is aided by an attention mechanism which implicitly learns which regions within the DWI are most relevant to the classification.

## 1 Introduction

Patients that have suffered the symptoms of a stroke have a very short time frame in which to be effectively treated; therefore, it is imperative that radiologists determine whether the cause of the symptoms is due to an ischemic lesion, hemorrage, or neither. Computer aided diagnostic (CAD) systems are commonly used by radiologists to assist in the interpretation 2D and 3D radiological images: from providing basic image processing, through to localisation and classification of pathologies. In the case of 3D diffusion weighted magnetic resonance images (DWI) of stroke patients, having a CAD system capable of flagging patients with potential lesions and extracting the relevant slices can greatly reduce the processing time of individual exams, as well as reduce the time-to-treatment for high-risk patients where time is critical.

A common approach to developing these systems is through supervised machine learning algorithms and large amounts of annotated data. Unlike in computer vision, high-quality annotated data requires the expertise of specialist radiologists, and it is incredibly time consuming to generate the amount required for a good predictive model. On the other hand, past radiological exams and corresponding free-text reports are available in large quantities and dispense with the need for manual labeling; however, their use in supervised learning present a different set of challenges. For one, the free-text reports detailed by clinicians are unstructured and may contain errors or omissions, and second, the language used, and even the image features chosen for description, can vary amongst clinicians. A single label can be extracted from the report by the presence/absence of a pathology, but would not capture the detail and context of the pathology within the image based on the description. For instance, a DWI clinician's report would typically detail the presence, location, severity and any visually descriptive features of the lesion.

1st Conference on Medical Imaging with Deep Learning (MIDL 2018), Amsterdam, The Netherlands.

To this end we propose an interpretable, attention-guided lesions localisation model that takes advantage of the descriptive clinical reports by training it to predict the pathology as well the brain region within which it is presented, as outlined in the clinical reports. Our learning framework consists of a recurrent attention model that implicitly learns the relevance of regions within individual slices of a DWI to the final diagnosis.

To our knowledge, this is the first attempt to use raw radiological reports gathered from past examination in a multi-modal attention-guided acute lesion prediction framework that takes multiple slices from a 3D image as input. In addition, this framework has been trained and tested on a real clinical dataset with real potential clinical applications as part of computer aided diagnostic system.

## 2  Related work

### 2.1  Learning from multi-modal data

Learning from natural language descriptions of images is primarily used towards the goal of image caption generation, the objective being complex scene understanding that goes beyond simple object recognition. In computer vision, the most sophisticated models use deep learning. Typically, recurrent neural networks (RNNs) are combined with convolutional neural networks (CNNs) and trained using backpropagation techniques [1, 2, 3]. Such models and learning frameworks are, to a lesser extent, being applied to medical images and their reports: from learning to automate medical subject heading (MeSH®) annotations for chest X-rays [4], to leveraging reports in a dual-attention framework to improve features used for classifying histopathology images and to provide interpretability to the classification [5, 6]. Creating these templated reports and manual heading annotations are both time consuming tasks that can only be done by qualified radiologists, so the training data available is usually limited. In addition, the learning frameworks are constrained to these templates, and so cannot be easily transferred to other imaging modalities.

Another approach to learning from radiological images and clinical reports is text mining the reports for diagnoses and assigning them as labels to the images to be used in classification [7] and weakly supervised localisation learning frameworks [8]. In these examples, a series of text processing techniques are applied to the reports for pathology extraction, including negation detection, and tools such as DNorm [9] and MetaMap [10], which map key words to a standardised vocabulary of clinical terms. However, other biological concepts in the reports, such as location, severity, visually descriptive features of the pathology, and concepts present in patient history, are not taken advantage of.

Here we propose to use pathologies and their locations extracted from raw textual reports from past radiological exams for the following reasons: locations can aid in pathology localisation and provide more context and interpretability to the output, we do not need to rely on radiologists for manual annotation of images and are therefore able to acquire a large number of training images, and the framework is not specific to an imaging modality.

### 2.2  Recurrent attention-guided supervision

Attention mechanisms have been successfully used in machine translation [11], image classification [12] and image captioning [2] in order to learn to attend to parts of the input: words in text, image regions, or both simultaneously. Attention is learned over image regions by computing a context vector $\mathbf{x_t} = \phi\left(\{\mathbf{a}_i\}, \{\alpha_i\}\right)$ which is a dynamic representation of the relevant parts of the image at time step $t$ for each location $i$, where $\alpha_i$ are the weights of each image feature vector $\mathbf{a}_i$. For single, 2D image classification, these annotation vectors are taken from a lower convolutional layer of a CNN. A recurrent neural network (RNN) processes these inputs at each time step, learning a sequential internal representation of locations based on the prediction task. At each time step, the weights $\alpha_i$ are computer for each location $i \in 1 \cdots L^2$ as per the formulation in [11] and [2]:

$$e_{ti} = \mathrm{MLP}\left(\mathbf{a}_i, \mathbf{h}_{t-1}\right) \quad \alpha_i = \frac{\exp\left(e_{ti}\right)}{\sum_i \exp\left(e_{ti}\right)} \tag{1}$$

where MLP is multi-layer perceptron, $\mathbf{h}_{t-1}$ is the hidden state of the RNN at the previous time step. The location $i$ for the next time step can be found by sampling from this softmax (hard attention), or

by computing the expectation over the feature slices (soft attention), the advantage of soft attention being that it is differentiable. In attention-guided multilabel video classification tasks [13, 14], the RNN is used to model the temporal dependencies of frames and attention is learnt over locations within individual frames. We adopt a similar approach where an RNN is used to model the sequential dependencies of slices within the DWI and trained to produce a multilabel output, where each label pathology/region within the brain as extracted from the reports.

## 3  Method

A Long Short-Term Memory (LSTM) [15] RNN is used to model the sequence of input slices. Each LSTM unit has three sigmoid gates to control the internal state: 'input', 'output' and 'forget'. At each time step, the gates control how much of the previous time steps is propagated through to determine the output. For an input sequence $\mathbf{X} = \{x_1, \ldots, x_N\}$, the internal hidden state $h_t$ and memory state $c_t$ are updated as follows:

$$
\begin{aligned}
\mathbf{i}_t &= \sigma(W^{(iy)}\mathbf{y}_{t-1} + W^{(ih)}\mathbf{h}_{t-1} + W^{(ix)}\mathbf{x}_t) \\
\mathbf{f}_t &= \sigma(W^{(fy)}\mathbf{y}_{t-1} + W^{(fh)}\mathbf{h}_{t-1} + W^{(fx)}\mathbf{x}_t) \\
\mathbf{o}_t &= \sigma(W^{(oy)}\mathbf{y}_{t-1} + W^{(oh)}\mathbf{h}_{t-1} + W^{(ox)}\mathbf{x}_t) \\
\mathbf{c}_t &= \mathbf{f}_t \odot \mathbf{c}_{t-1} + \mathbf{i}_t \odot \tanh(W^{(cx)}\mathbf{x}_t + W^{(ch)}\mathbf{h}_{t-1}) \\
\mathbf{h}_t &= \mathbf{o}_t \odot \tanh(\mathbf{c}_t)
\end{aligned}
\tag{2}
$$

where $W^{(cx)}$ and $W^{(ch)}$ are the trainable weight parameters, and $\mathbf{i}_t$, $\mathbf{o}_t$ and $\mathbf{f}_t$ are the input, output and forget gates respectively.

For each 3D DWI, the annotations vectors of each slice are taken from the last convolution layer of a CNN: $a = \{\mathbf{a}_1, \ldots, \mathbf{a}_N\}, \mathbf{a}_i \in \mathbb{R}^{L \times L \times D}$. The weights can be thought of as the probability distribution of the relevancy of each location to the output (diagnosis). The input at the next time step $x_t$ is then the expectation of features at different locations:

$$
x_t = \sum_i^{L^2} \alpha_i \mathbf{X}_i
\tag{3}
$$

We train by learning to predict the output $y_t$, which is a k-hot vector of labels summarising the report. The complete model is illustrated in Figure 1.

## 4  Experiments

### 4.1  Data

The dataset used in this study consisted of 1226 DWI scans and corresponding radiological reports of acute stroke patients, collected from local hospitals. All the images and reports were fully anonymised and ethical approval was granted by Imperial College Joint Regulatory Office. The images varied in sizes between $(7–52) \times (64 \times 64) – (512 \times 512)$, with slice thickness: 5mm, slice spacing: 1.0–1.5mm, and pixel size in x–y plane: $1.40 \times 1.40 – 1.80 \times 1.80$. The scans were pre-processed according to the steps outlined in [16]: images were resampled into uniform pixel size of $1.6 \times 1.6$mm, and pixel intensities were normalised to zero mean and unit variance. The images were then re-scaled and padded to $128 \times 128$.

Each report was parsed by a clinician to extract 1–2 sentences summarising the presence/absence of the pathology and its location within the brain. These filtered reports contained between 1 and 78 words, with an average of 16.7 and standard deviation 9.8. In addition, each exam is assigned a diagnosis label as part of hospital protocol: 54% were diagnosed 'no acute infarct', 46% were diagnosed 'acute infarct'. The remaining, which made up a total of <1% and included diagnoses such as 'unknown', 'haematoma', 'tumour', were removed for the purpose of training, leaving a total of 1177 exams.

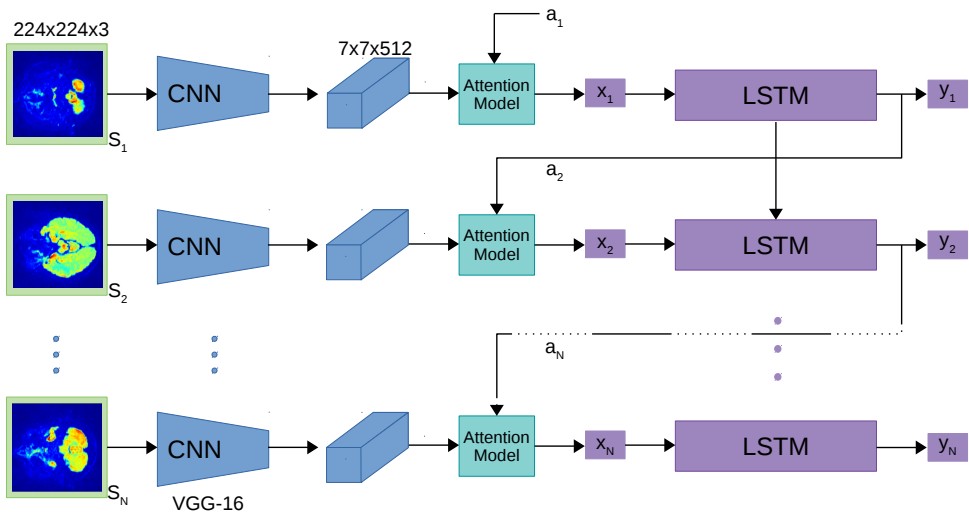

Figure 1: Attention-guided clinical report generation model.

Table 1: Top 20 classes of brain regions after re-assignment based on a hierarchical ontology.

| Brain region | freq. | Brain region | freq. |
|---|---|---|---|
| frontal lobe | 100 | left cerebellar hemisphere | 17 |
| basal ganglia | 99 | cerebellum | 15 |
| parietal lobe | 74 | centrum semiovale | 13 |
| corona radiata | 72 | posterior cerebral artery | 12 |
| middle cerebral artery | 68 | medulla oblongata | 11 |
| occipital lobe | 45 | midbrain | 11 |
| pons | 43 | superior cerebellar artery | 11 |
| insular cortex | 41 | perirolandic region | 9 |
| thalamus | 39 | thalamocapsular region | 8 |
| temporal lobe | 36 | right cerebellar hemisphere | 8 |

## 4.2 Brain region extraction

A combination of manual and automated annotation was necessary in order to extract terms relating to brain regions from the clinical reports. A hierarchical brain region ontology available from the Allen Institute [17] was used to manually extract the terms, and then automatically assign these terms to larger, parent regions in the hierarchy. For instance, 'left middle temporal gyrus' is located within, and therefore reassigned to, the 'temporal lobe'. Ones that occurred less than 3 times were excluded. In this way, 356 unique regions were reduced to 42. The 20 most common regions and their frequency of appearance are listed in Table 1. The diagnosis (binary presence/absence of acute infarct) and each region is treated as a class, and each report is thus encoded as a 43-hot vector. An example of a DWI central slice, its corresponding clinical report, manually extracted regions and re-assigned region labels is illustrated in Figure 2.

## 4.3 Model and training parameters

We evaluate the effectiveness of the attention-guided model by comparing it to basic fine-tuning of VGG-16 [18] and GoogleNet [19], pretrained on the ImageNet [20] dataset, for multilabel classification. For the fine-tuning, the weights of the CNN are held fixed up to the last average pooling layer ($\mathbb{R}^{4096}$ for VGG-16, $\mathbb{R}^{1024}$ for GoogleNet), compared to the last convolutional layer of the

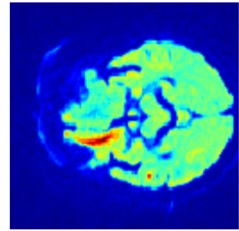

**Clinical Report:** There is a small focus of acute ischaemia in the right corona radiate, and a tiny focal cortical infarct in the left middle temporal gyrus.

**Clinical Diagnosis:** Acute infarct

**Manually Extracted Regions:** right corona radiate, left middle temporal gyrus

**Report Labels:** acute_infarct, corona_radiata, temporal_lobe.

Figure 2: Central slice of sample DWI exam with corresponding clinical report, clinical diagnosis, manually extracted brain regions and labels.

attention framework ($\mathbb{R}^{14 \times 14 \times 512}$ for VGG-16). Since lesions may be located in any slice(s) within the DWI, we first run fine-tuning by taking the central slice as input, and then compare that to taking a max-aggregate of pooling features across all slices.

The VGG-16 and GoogleNet models take a fixed-size input of $224 \times 224 \times 3$, each slice is padded, and duplicated. The dimension of the hidden state of the LSTM is set to $\mathbb{R}^{512}$. The LSTM is unrolled up to 19 time steps for the average number of slices. Images with fewer than 19 slices were re-distributed and padded with intervening slices. The LSTM model was trained by minimising the cross-entropy loss:

$$L(S, I) = -\sum_{t=0}^{K} \sum_{c=0}^{C} y_{t,c} \log \hat{y}_{t,c} + \lambda \sum_{i} W_i^2 \tag{4}$$

where $y_t$ is the k-hot vector of labels at time step $t$ and $N$ is the LSTM sequence length, $\lambda$ is the weight decay coefficient, and $W$ are all the model parameters. Exams were split into 80%-10%-10% for training, testing and validation respectively. At training time, loss is minimised over the training set using stochastic gradient descent (batch size 16, learning rate 1e-5, 10 epochs), and parameters are updated using Adam [21] optimisation.

## 5 Results

To test the effectiveness of learning attention over image slices, we evaluate the mean average accuracy, precision and recall as outlined in [22], and Hamming Loss as outlined in [23], comparing the results to standard VGG-16 and GoogleNet models trained without attention. Table 5 summarises the quantitative results. We report the accuracy of the 'infarct' class separately as it is ultimately the one that radiologists are interested in, and we also report the mean average accuracy, precision and recall (mAA, mAP, mAR) and Hamming Loss (HL) of all the classes.

Taking the central slice as input and fine-tuning performs better than taking all of the slices and max-aggregating the features. This is as expected since lesions will only be present in a small number of slices, and taking all the slices as inputs introduces a lot of noise. On the other hand, taking the central slice is the more naive approach: a lesion reported in the text may not be present in the central slice. Taking an expectation over all the locations across the slices provides a compromise: the entire image is explored, but the input is more localised at each time step. The accuracy of the 'infarct' class has improved over the standard models, and in addition, an improvement is seen in mean average precision, which is arguably more important in the medical domain as we want to identify all patients with high-risk lesions (for further examination).

### 5.1 Qualitative assessment of attention over image slices

Figure 3 displays a number of sample DWI images that have been classified by the model as containing an 'acute infarct'. They are presented alongside their corresponding true and predicted labels, as well as attention maps overlaid over each slice to give a sense of which parts of the image slices were used to determine the output labels. The attention maps inform us as to whether the model is looking over the correct areas of the brain, and whether it is 'exploring' the full 3D image in order to make a classification. We can see that the model is looking at a combination of various localised regions within the slices of the DWI in order to make the classification, focusing particularly on regions of

Table 2: Classification accuracy of visual and textual recurrent attention models. The attention model is compared to VGG-16 and GoogleNet pretrained on ImageNet and fine-tuned for multilabel classification. The accuracy of the 'infarct' class is reported separately as well as part of the mean average accuracy, precision and recall (mAA, mAP, mAR) and Hamming Loss (HL) of all classes.

| | Acc. 'infarct' (%) | mAA (%) | mAP (%) | mAR (%) | HL (%) |
|---|---|---|---|---|---|
| GoogleNet, central slice | 62.7 | 97.5 | 26.7 | 14.0 | 2.5 |
| VGG-16, central slice | 59.3 | 97.3 | 29.3 | 17.1 | 2.7 |
| VGG-16, max-aggregate | 55.1 | **97.9** | 24.6 | 14.2 | **3.1** |
| VGG-16+attention | **68.0** | 97.7 | **39.0** | **19.6** | 2.2 |

high contrast. Acute stroke lesions appear as regions of hyperintensity in DWI images, however, we would need the expertise of radiologists to make a more quantitative assessment.

## 6 Conclusion

We present a first attempt at a multi-modal lesion detection framework that uses an attention mechanism in order to better learn to predict the presence of lesions within 3D DWI, with the help of brain region locations extracted from textual reports. We show that learning implicit attention over the 3D image for multilabel classification results in better overall performance when compared to standard approaches of fine-tuning. The next steps are to use other visually descriptive entities in the reports, such as 'several small foci', 'large volume' as these provide further context to the pathology, and can be used to not only improve the attention model, but also as part of a CAD system that can output a structured report at the end.

**Acknowledgments.**

This work is supported by the NIHR Grant i4i: Decision-assist software for management of acute ischemic stroke using brain-imaging machine-learning (Ref: II-LA-0814-20007).

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
