# OpenReview forum: "Learning Attention from Multi-Modal Imaging and Text: Application for Lesion Localisation in DWI"
_MIDL.amsterdam/2018/Conference — Submitted to MIDL 2018_

### Review · AnonReviewer2 · 2018-05-07
**Interesting idea and relevant question but the work might be a bit premature**

**Rating:** 2
**Confidence:** 3

**Review:**

The article aims to address the lesion localisation question by using radiology reports as labels rather than pixel-wise annotations often used for such work. However, authors end up addressing image-wide classification problem instead.

Pros:
1. The question is very relevant and an exciting one. It is indeed important to be able to use radiology reports as label information in order to make use of currently dormant large existing images.
2. Using attention networks is a good idea for this task.

Cons: Despite its positive sides, the presented article raises a number of concerns.
1. Authors start by aiming to solve the localisation problem but do not in the end solve it. The maps shown in Figure 3 are not necessarily indicating where the lesions are.

2. As a result, authors solve image-wide classification problem for which many methods exist. Authors use fine-tuning of existing recognition networks. However, the way they are implemented is not very convincing. Authors either take the central slide or aggregate slides together to reach a volume-wide classification. Method of aggregation is not very clear but I suspect it is not learned. Learning to aggregate over slides would probably be a reasonable approach

3. The method is not very well described. For instance,
3a. in section 2.2 authors introduce the time variable, which is confusing.
3b. In the experiments, it is not clear what a 43-hot vector is.
3c. Do all the slices get the same y labels or do labels vary across slices?
3d. The explanation for how standards methods are adapted to work with DWI data is not explained in detail.

The presented article is on an exciting research topic and a very relevant problem in medical image analysis. However, I find the submitted article premature.

**Special Issue:**

No

---

### Review · AnonReviewer3 · 2018-05-09
**A good paper**

**Rating:** 4
**Confidence:** 3

**Review:**

Overall:
The paper proposes an attention-based multi-modal deep learning model to provide more accurate medical diagnosis. Additionally, the attention mechanism allows to highlight Regions of Interests in an medical image that could be further used to assist a clinician in her daily work. The paper is well-written and all methods are properly selected. The results are sound.

Strengths:
+ The paper tackles a very important problem of the explainable AI in application to medical imaging. In order to allow a wide applicability of AI in medical imaging, providing an explanation and/or visual interpretation is a crucial step towards achieving this long-standing goal.
+ The proposed approach is sound and appropriate.
+ The experiments are well-performed with current state-of-the-art methods.
+ The paper has a vast of possible extensions and interesting future directions.

Remarks:
* Major
- On page 3, the authors say that: "The images were then re-scaled and padded to 128 × 128". However, later on page 5 and in Figure 1 they stress out that the VGG-16 or GoogleNet require 224x224x3 images. Could you explain this difference in sizes?
- The caption of Figure 1 states "Attention-guided clinical report generation model". This statement is false and misleading because the proposed model does not allow to generate a textual report. Please re-phrase this caption.
- How many times were the experiments run? Some differences in Table 2 are very small and I wonder whether they are within one standard error.

* Minor
- A closer inspection of the attention maps in Figure 3 reveals that the highlighted regions are not necessarily useful for the visual explanation. My impression is that in most cases the attention mechanism just highlights a region of an image that is occupied by a brain without any particular regions. Is it really the case or maybe the idea is not fully presented in Figure 3?
- I agree with the authors that the proposed approach is probably the first attempt to explain a diagnosis using a text report together with visualizing ROI in the medical imaging domain, however, such attempts were successfully done in computer vision, e.g.:
o Park, D. H., Hendricks, L. A., Akata, Z., Rohrbach, A., Schiele, B., Darrell, T., & Rohrbach, M. (2018). Multimodal Explanations: Justifying Decisions and Pointing to the Evidence. arXiv preprint arXiv:1802.08129.

It would be beneficiary to discuss this approach and compare it to the proposed method. Moreover, I miss some relevant literature on explainable AI using images are text, such as:
o Hendricks, L. A., Akata, Z., Rohrbach, M., Donahue, J., Schiele, B., & Darrell, T. (2016, October). Generating visual explanations. In European Conference on Computer Vision (pp. 3-19). Springer, Cham.
o Hendricks, L. A., Hu, R., Darrell, T., & Akata, Z. (2017). Grounding Visual Explanations. arXiv preprint arXiv:1711.06465.

and the attention mechanism:
o Li, K., Wu, Z., Peng, K. C., Ernst, J., & Fu, Y. (2018). Tell Me Where to Look: Guided Attention Inference Network. arXiv preprint arXiv:1802.10171.
o Ilse, M., Tomczak, J. M., & Welling, M. (2018). Attention-based deep multiple instance learning. arXiv preprint arXiv:1802.04712.


**Special Issue:**

No

---

### Review · AnonReviewer1 · 2018-05-12
**Attention architecture not convincing, text is manually translated to a binary feature vector. Localisation claim is not supported by experiments**

**Rating:** 1
**Confidence:** 3

**Review:**

The authors train a network to predict a binary "acute stroke" label and 42 binary anatomical region labels from a Diffusion Weighted Imaging (DWI) scan. The labels were manually extracted from radiologists reports. Their proposed architecture applies pre-trained and fine-tuned  2D VGG networks to every slice and run an LSTM with attention over the extracted slice-wise feature vectors, re-interpreting the z-dimension as temporal dimension.
Unfortunately the motivation for this architecture is not explained. I.e. why do the authors expect that the first (the topmost slice if they iterate top-down) contain the information for an attention map in the second slice? Why limit the context for each slice to only the one side?
A very simple (presumably more powerful) baseline would be to feed the slice-wise features into a CNN instead an RNN. Of course a 3D CNN would be even more powerful.
Beside that, the claim that the approach uses text seems a little bit misplaced, given the fact that the text was manually translated to a binary feature vector. Furthermore the authors do not present an experiment that supports the claim that their approach can localise lesions.



**Special Issue:**

No

---

### Decision · Program_Chairs · 2018-05-15
**Paper14 Acceptance Decision**

Reject